# Biological Effects of Intravenous Vitamin C on Neutrophil Extracellular Traps and the Endothelial Glycocalyx in Patients with Sepsis-Induced ARDS

**DOI:** 10.3390/nu14204415

**Published:** 2022-10-21

**Authors:** Xian Qiao, Markos G. Kashiouris, Michael L’Heureux, Bernard J. Fisher, Stefan W. Leichtle, Jonathon D. Truwit, Rahul Nanchal, Robert Duncan Hite, Peter E. Morris, Greg S. Martin, Jonathan Sevransky, Alpha A. Fowler

**Affiliations:** 1Division of Pulmonary Disease & Critical Care Medicine, Virginia Commonwealth University, Richmond, VA 23219, USA; 2Division of Acute Care Surgical Services, Virginia Commonwealth University, Richmond, VA 23219, USA; 3Division of Pulmonary Disease & Critical Care Medicine, Froedtert and the Medical College of Wisconsin, Aurora St. Luke’s Medical Center, Milwaukee, WI 53215, USA; 4Division of Pulmonary Disease & Critical Care Medicine, University of Cincinnati, Cincinnati, OH 45221, USA; 5Division of Pulmonary Disease & Critical Care Medicine, University of Alabama at Birmingham, Birmingham, AL 35294, USA; 6Division of Pulmonary Disease & Critical Care Medicine, Emory University, Atlanta, GA 30322, USA

**Keywords:** sepsis, vitamin C, glycocalyx, syndecan-1, cell-free DNA, acute respiratory distress syndrome

## Abstract

(1) Background: The disease-modifying mechanisms of high-dose intravenous vitamin C (HDIVC) in sepsis induced acute respiratory distress syndrome (ARDS) is unclear. (2) Methods: We performed a post hoc study of plasma biomarkers from subjects enrolled in the randomized placebo-controlled trial CITRIS-ALI. We explored the effects of HDIVC on cell-free DNA (cfDNA) and syndecan-1, surrogates for neutrophil extracellular trap (NET) formation and degradation of the endothelial glycocalyx, respectively. (3) Results: In 167 study subjects, baseline cfDNA levels in HDIVC (84 subjects) and placebo (83 subjects) were 2.18 ng/µL (SD 4.20 ng/µL) and 2.65 ng/µL (SD 3.87 ng/µL), respectively, *p* = 0.45. At 48-h, the cfDNA reduction was 1.02 ng/µL greater in HDIVC than placebo, *p* = 0.05. Mean baseline syndecan-1 levels in HDIVC and placebo were 9.49 ng/mL (SD 5.57 ng/mL) and 10.83 ng/mL (SD 5.95 ng/mL), respectively, *p* = 0.14. At 48 h, placebo subjects exhibited a 1.53 ng/mL (95% CI, 0.96 to 2.11) increase in syndecan-1 vs. 0.75 ng/mL (95% CI, 0.21 to 1.29, *p* = 0.05), in HDIVC subjects. (4) Conclusions: HDIVC infusion attenuated cell-free DNA and syndecan-1, biomarkers associated with sepsis-induced ARDS. Improvement of these biomarkers suggests amelioration of NETosis and shedding of the vascular endothelial glycocalyx, respectively.

## 1. Introduction

Acute respiratory distress syndrome (ARDS) is an inflammatory disease of the lungs with a mortality rate of 35% to 46% [1,2,3,4]. Sepsis can trigger vascular injury that leads to ARDS through systemic and local inflammation, damaging lung barrier function through both alveolar and capillary injury. Loss of lung barrier integrity leads to interstitial and alveolar flooding, surfactant damage, and collapsed lung units. The cumulative result is severe pulmonary function impairment, diminished lung compliance, increased shunt, and hypoxemia.

Clinical and laboratory evidence suggests that high-dose intravenous vitamin C (HDIVC) may play a role in the treatment of ARDS [5,6,7,8,9]. A recently completed randomized, double-blind, placebo-controlled, multicenter trial suggested that HDIVC had an association with reduced 28-day all-cause mortality in sepsis-associated ARDS [8]. We hypothesized that the key mechanisms of action of HDIVC pertain to the preservation of the endothelial glycocalyx and the regulation of neutrophil extracellular trap (NET) formation [10,11,12].

A key biomarker for glycocalyceal integrity is the proteoglycan syndecan-1, an important structural component of the glycocalyx that lines luminal endothelial surfaces of the vasculature, including alveolar capillaries [13]. Endotoxemia and bacteremia that lead to sepsis disrupt the glycocalyx, one of the earliest and most significant sites of injury [14]. Early phase injury leads to degradation of the glycocalyx barrier and shedding of syndecan-1 into the circulation [15]. High plasma syndecan-1 levels are associated with the development of ARDS [10,16]. Loss of glycocalyceal integrity results in movement of proteins and fluid from the vasculature into the perivascular interstitial space, a hallmark of ARDS. Therapies that target protection or restoration of the glycocalyx may benefit septic subjects and possibly reduce ARDS-associated mortality [17].

NETs, a recent discovery in innate immunity, are composed of granular content and nuclear content used to kill bacteria extracellularly [18]. A key biomarker of NET formation is cell-free DNA (cfDNA), which is elevated in sepsis and ARDS [11,18,19,20,21]. Prior studies have shown higher levels of cfDNA and syndecan-1 at the onset of sepsis and ARDS as predictors of increased level of sepsis, mortality rate, and likelihood of intubation [10,16,22,23]. To date, no study has investigated an intervention that reduces NET formation and glycocalyx degradation. We hypothesized that HDIVC could reduce plasma cfDNA and syndecan-1 levels of subjects with sepsis-induced ARDS, compared to subjects receiving standard of care. We further hypothesized that cfDNA and syndecan-1 correlate with objective clinical oxygenation indices in subjects with ARDS.

## 2. Materials and Methods

We performed a post hoc analysis of plasma biomarkers obtained from subjects enrolled in CITRIS-ALI, a recently completed multicenter, double-blind, randomized, placebo-controlled trial [9]. In this trial, critically ill subjects with new-onset sepsis-induced ARDS, were randomized to receive 4-day bolus infusions of ascorbic acid (*n* = 84) (McGuff Pharmaceuticals, Santa Ana, CA, USA) at 50 mg/kg every 6 h, for 96 h vs. placebo (*n* = 83).

Whole blood was drawn into sterile Vacutainer tubes (BD 367863, lavender top, K2EDTA). Plasma was separated by centrifugation (1000× *g*, 10 min, 4 °C), aliquoted and frozen at −80 °C for batch analysis. Plasma cfDNA levels were blindly prepared and quantified using the Invitrogen Quant-iT PicoGreen dsDNA assay kit, according to the manufacturer’s instructions (Thermo Fisher Scientific, Waltham, MA, USA). Fluorescence intensity was measured on a SpectraMax Gemini XPS microplate reader with excitation at 490 nm and emission at 525 nm, with 515 nm emission cutoff filter (Molecular Devices, San Jose, CA, USA). Plasma syndecan-1 levels were blindly prepared and analyzed using a human magnetic bead Luminex assay system (LXSAHM), according to the manufacturer’s instructions (R&D Systems, Minneapolis, MN, USA) and quantified using a Luminex LX200 instrument with xPONENT 3.1 software (Luminex Corporation, Austin, TX, USA). Biomarker concentrations were calculated from standard curves of Median Fluorescence Intensity (MFI) by generating a five-parameter logistic (5-PL) curve-fit and multiplying by the dilution factor. Specimens outside the standard range were further diluted and the assay repeated. Samples analyzed were from time points 0 h and 48 h due to maximal specimen availability.

Analyses were conducted using Stata Statistical Software (Rel.16.0, TX StataCorp LP, College Station, TX, USA). We applied descriptive statistics to report the baseline characteristics of the study population. We applied linear regressions to evaluate the association of the 48 h change of the biomarkers [24,25]. Multiple linear regressions were applied to assess biomarker differences at 48 h, adjusting for baseline biomarker levels. The researchers chose the 48 h mark instead of the 96 h because the 96 h endpoint had less specimens that 48 h mark. This was a result of survival bias in the HDIVC arms, as most of the mortality difference between the two arms occurred within 48 h treatment initiation. Post-estimation scatter plots were constructed to graphically represent the findings in the two groups. Regression residuals for normalcy were then assessed. Multiple logistic regression was applied to evaluate the adjusted effect of biomarkers on mortality. We evaluated the models with the area under the receiver operator characteristic curve, and the Hosmer-Lemeshow goodness-of-fit test. The funding sponsors had no role in the biomarker study described here. The corresponding author had full access to all data in the study and had final responsibility for the decision to submit for publication.

## 3. Results

### 3.1. Study Participants

Subjects with sepsis-induced ARDS were enrolled at the time of ARDS onset (*n* = 167). Subjects were randomized to receive HDIVC in dextrose 5% in water (*n* = 84) or placebo (dextrose 5% in water, *n* = 83). Baseline plasma samples were analyzed for cfDNA (*n* = 167) and syndecan-1 (*n* = 166). At 48 h, 82 (97.6%) HDIVC subjects and 72 (86.8%) of the placebo subjects survived and remained in ICU (*p* = 0.009, Pearson Chi-Square). Plasma specimens from the survivors were analyzed. Details on study subjects and specimens can be found in Table 1.

### 3.2. Effects of Intravenous Vitamin C on Plasma cfDNA

At baseline, mean cfDNA levels in HDIVC treatment and placebo arms were 2.18 ng/µL (SD 4.20 ng/µL) and 2.65 ng/µL (SD 3.87 ng/µL), respectively, and there was no statistical difference (*p* = 0.46). After 48 h, mean cfDNA levels in HDIVC and placebo arms were 1.78 ng/µL (SD 1.73 ng/µL) and 2.80 ng/µL (SD 5.0 ng/µL), respectively. The mean change (delta, Δ) following 48 h of HDIVC treatment was −0.45 ng/µL (95% CI, −1.16 to 0.25), indicating a decrease in cfDNA levels. The mean change following 48 h in placebo subjects increased by 0.57 ng/µL (95% CI, −0.19 to 1.33 ng/µL). Adjusting for baseline cfDNA levels, subjects receiving HDIVC treatment exhibited a mean 48 h cfDNA decrease of 1.02 ng/µL (*p* = 0.05) compared to placebo (Figure 1).

### 3.3. Effects of Intravenous Vitamin C on Plasma Syndecan-1

Mean baseline syndecan-1 levels in HDIVC and placebo arms were 9.49 ng/mL (SD 5.57 ng/mL) and 10.83 ng/mL (SD 5.95 ng/mL), respectively, with no statistical difference (*p* = 0.14). At 48 h, syndecan-1 levels in HDIVC subjects increased to 10.43 ng/mL (SD 6.05 ng/mL) while corresponding levels in placebo subjects increased to 11.22 ng/mL (SD 5.31 ng/mL). The change in placebo subjects at 48 h (1.53 ng/mL, 95% CI, 0.96 to 2.11) was twice that of HDIVC subjects (0.75 ng/mL, 95% CI, 0.21 to 1.29, *p* = 0.05). (Figure 2).

### 3.4. Effects of Plasma Syndecan-1 on 28-Day All-Cause Hospital Mortality

Baseline syndecan-1 levels as well as both 48 h changes in cfDNA and syndecan-1 levels predicted 28-day all-cause mortality. Table 2 outlines the Odds Ratio (OR) of death for every incremental unit of the corresponding biomarker.

Syndecan-1 levels at 0 and 48 h correlated significantly with increased plasma cfDNA levels at 0 and 48 h, respectively (Figure 3a,b). Furthermore, the 48 h change of plasma syndecan-1 levels correlated with the 48 h change of coda levels (Figure 3c).

### 3.5. Effect of cfDNA and Syndecan-1 on Changes in Oxygenation

Forty-eight-hour syndecan-1 plasma level elevations were significantly correlated with worsened oxygenation. The incremental increase in syndecan-1 level by one ng/mL corresponded with a change in PaO2/FiO2 ratios of −8.85 (95% CI, −17.50 to −0.19; *p* = 0.045). HDIVC treatment had a larger (−18.9 vs. −4.4) and significant (*p* = 0.004 vs. 0.48) impact on ΔPaO2/FiO2 ratio compared with placebo (Figure 4). The 48 h ΔcfDNA exhibited no significant effects on PaO2/FiO2 ratios (−2.72 [95% CI, −9.66 to 4.20], *p* = 0.44).

## 4. Discussion

The present study reports that a 48 h HDIVC infusion in subjects with sepsis-associated ARDS attenuated increases in 48 h cfDNA and syndecan-1 plasma levels. Attenuated syndecan-1 levels correlated with improved lung function, as gauged by improved 48 h PaO2/FiO2 ratios (Figure 4). HDIVC’s impact on syndecan-1 and cfDNA levels independently predicted lower 28-day all-cause mortality.

Syndecan-1 elevations at baseline at time of randomization for HDIVC vs. placebo and increases at 48 h compared to baseline increased the 28-day all-cause mortality odds ratio. For each increase of one ng/mL of syndecan-1, there was an associated odds ratio increase of 1.3 for 28-day all-cause mortality, at baseline and the delta at 48 h. Elevated cfDNA at baseline did not show a significant effect on mortality. However, each increase in cfDNA by one ng/µL at 48 h compared to time of randomization for HDIVC vs. placebo, there was associated odds ratio increase of 1.3 for 28-day all-cause mortality (Table 2). As HDIVC directly attenuates the rise of both syndecan-1 and cfDNA at 48-h, this may provide a partial pathway into the mechanism of mortality benefit with HDIVC in subjects with sepsis induced ARDS (Figure 1 and Figure 2). Furthermore, cfDNA and syndecan-1 levels in the plasma of septic subjects with ARDS provides fresh insight into the extent of systemic inflammation and the molecular mechanisms that produce vascular injury, leading to ARDS onset.

Neutrophil extracellular traps (NETs) are highly linked to endothelial damage and organ failure, crucial events in sepsis [19]. NET formation is a neutrophil effector mechanism whereby neutrophils extrude a web of chromatin fibers complexed to granule-derived antimicrobial peptides and enzymes. This process occurs following neutrophil activation and is implicated in producing endothelial damage [26]. Hirose et al. identified NETs in peripheral blood smears of critically ill subjects [27]. In septic subjects, circulating cfDNA levels correlated with the degree of lung injury, as higher concentrations were found in subjects who developed moderate or severe ARDS than septic subjects without ARDS [28]. LeFrancais et al. found that attenuating NET formation in an acute lung injury mouse model led to increased survival [21]. Activated endothelial cells induce neutrophil NET formation and are themselves susceptible to NETosis-mediated cell death, [29] thus, promoting a self-perpetuating damage that ultimately leads to hypercoagulable states [30]. The association of syndecan-1 with endothelial damage and neutrophilic inflammation has focused attention on a biomarker indicative of vascular injury [15,17,31]. Plasma syndecan-1 levels in septic subjects are increased at baseline and may remain elevated for up to 72 h [32]. Further, in these septic subjects, elevations of syndecan-1 are associated with heightened risks of developing respiratory failure and increased mortality [33]. Both cfDNA and syndecan-1 levels are reported for mortality predictions in septic subjects and are associated with adverse clinical outcomes (e.g., development of multiple organ failure, ARDS) [14,34]. Correlations between the two biomarkers to clinical outcomes pertain to their roles as surrogates for NET formation and glycocalyx integrity. Plasma syndecan-1 elevations are a robust marker of glycocalyx degradation and development of ARDS [31]. To our knowledge, this is the first human randomized placebo-controlled study of sepsis-associated ARDS to examine an interventional therapy’s effect on these biomarkers.

CITRIS-ALI is the first study to show that a 96 h infusion of HDIVC decreased human plasma cfDNA [8]. This post hoc analysis of the CITRIS-ALI study shows HDIVC attenuated the rise in syndecan-1 levels at 48 h (Figure 1 and Figure 2). A re-analysis of the CITRIS-ALI data, accounting for the missing SOFA scores due to the large survival differences among the two arms, (i.e., Survivorship Bias), demonstrated improved overall organ-function (modified SOFA Scores) in ARDS patient who received HDIVC infusion [9]. Murine models of polymicrobial sepsis using high dose vitamin C have demonstrated attenuation of lung NET formation and circulating cfDNA [5]. Other studies reveal that high dose vitamin C reduced multiple organ failure, neutrophilic capillaritis, and extravascular lung water in septic mice [6]. Thus, decreased circulating cfDNA may represent a surrogate marker of high dose vitamin C’s ability to reduce excessive neutrophilic activation during sepsis, thereby reducing NET formation and the ensuing inflammatory vascular injury. The present study found significant correlations between decreased plasma syndecan-1 and improved PaO2/FiO2 ratios (Figure 4) and 28-day all-cause hospital mortality (Table 1). Taken together, these findings suggest that a 96 h infusion HDIVC may improve ARDS recovery by protecting or restoring glycocalyx integrity and reducing neutrophil NET formation.

The study has several limitations. First, the specific origin of circulating cfDNA was not determined. One study suggests that surges in cfDNA in sepsis results from cellular necrosis [35]. However, other studies show that cfDNA in septic subjects is host derived and the cfDNA base pair length is consistent with neutrophil NET formation and not cellular necrosis [36]. Second, syndecan-1 was measured, but not other glycocalyceal structures (i.e., endocan, heparan sulfate, hyaluronan). Multiple studies show that syndecan-1 levels in sepsis correlate strongly with other markers of glycocalyx degradation [14,15]. Third, missing data points at 48 h due to early deaths in placebo subjects (13.3%) vs. HDIVC subjects (2.4%) may have biased the ability to detect an even greater difference in biomarker levels.

## 5. Conclusions

HDIVC treatment reduced 48 h cfDNA and syndecan-1 plasma levels in subjects with sepsis-associated ARDS. The dynamic changes of these biomarkers were strongly associated with lung oxygenation and 28-day all-cause mortality. These results suggest that HDIVC reduces the severity of illness by decreasing excessive NET formation and glycocalyx degradation. Syndecan-1 and cfDNA signal pathophysiological processes that lead to vascular injury in sepsis-associated ARDS. Future studies will clarify the role of these biomarkers in directing the care of subjects with sepsis induced ARDS.

## Figures and Tables

**Figure 1 nutrients-14-04415-f001:**
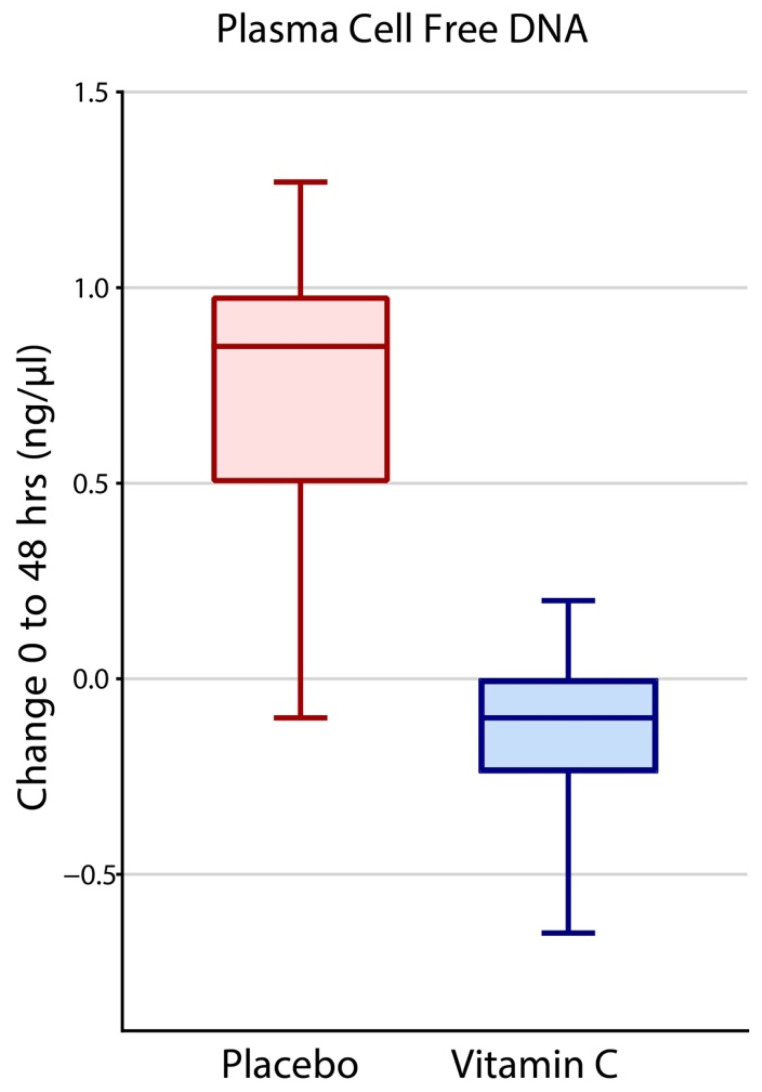
Forty-eight-hour change in cell-free DNA (cfDNA) in the two groups, Placebo and HDIVC. Median values, the top and bottom of the boxes show the interquartile range (IQR), whiskers show 95% CI. Outliers outside the 95% confidence intervals were graphically omitted.

**Figure 2 nutrients-14-04415-f002:**
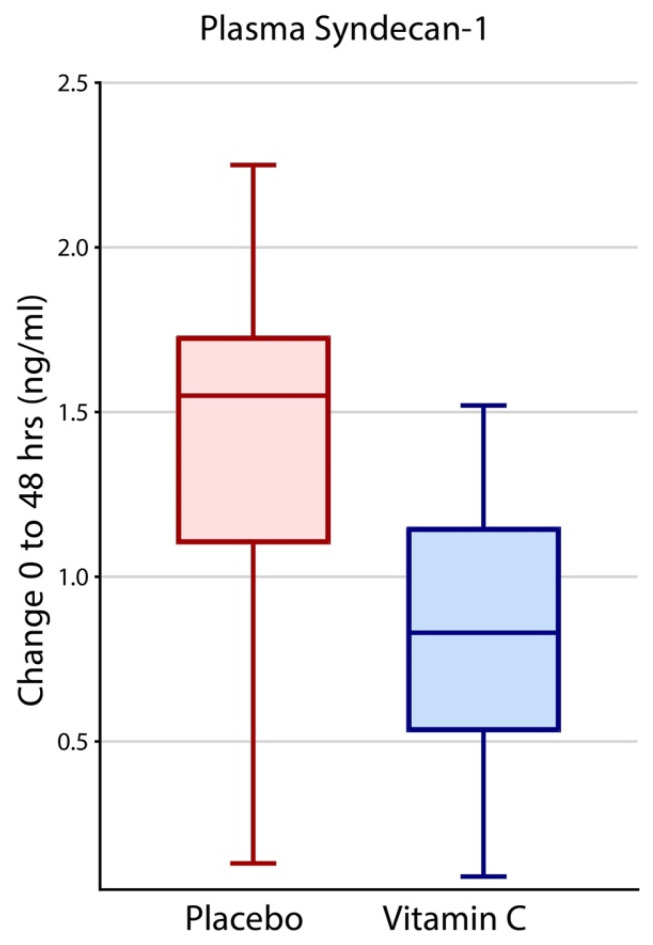
Forty-eight-hour increase in syndecan-1 in the two groups, Placebo and HDIVC. Median values, the top and bottom of the boxes show the interquartile range (IQR), whiskers show 95% confidence interval (CI). The extreme outliers outside the 95% confidence intervals have been omitted.

**Figure 3 nutrients-14-04415-f003:**
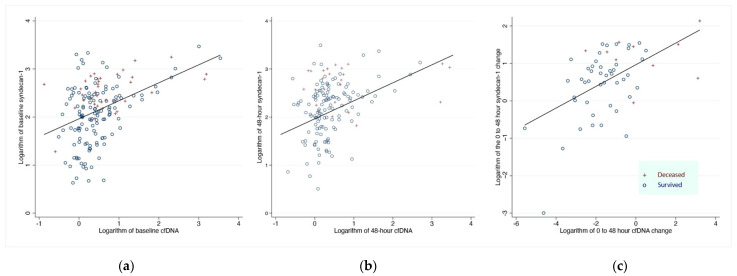
Scatterplot of (**a**) baseline levels and (**b**) 48 h levels and (**c**) 48 h change from baseline, of plasma cfDNA and syndecan-1. The fitted line corresponds to the regression line. Survivors at 7 days are represented with hollow circles, and expired subjects are represented with red crosses (+). Baseline (0 h): Beta = 0.37 *p* < 0.001 R-Squared = 0.18/48-h: Beta = 0.29, *p* < 0.001, R-Squared = 0.12/48 h change: Beta = 0.29 *p* < 0.001 R-Squared = 0.26. Abbreviations: cfDNA, cell-free deoxyribonucleic acid.

**Figure 4 nutrients-14-04415-f004:**
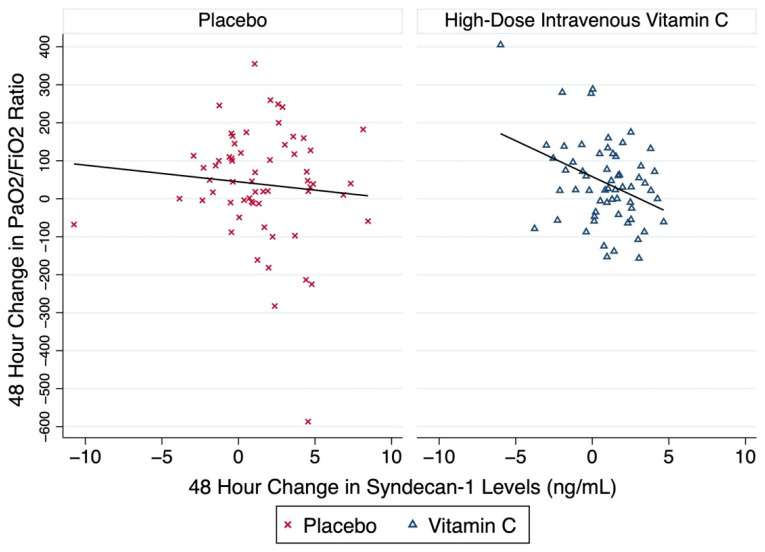
Scatterplots of the 48 h change in the PaO2/FiO2 ratio and the change in the plasma syndecan-1 during the same time-period. The figure illustrates the change of the PaO2/FiO2 ratio, a marker of oxygenation and lung injury which corresponds to the change in the plasma biomarker syndecan-1 among the two groups, Placebo and HDIVC. The linear fitted line corresponds to the regression line for each group: Placebo and HDIVC, respectively. The subjects of the placebo group are represented with red x symbols, and the subjects of the HDIVC group are represented with blue triangles. Abbreviations: FiO2, fraction of inspired oxygen; HDIVC, high-dose intravenous vitamin C; PaO2, arterial oxygen pressure.

**Table 1 nutrients-14-04415-t001:** Patient characteristics and specimen numbers from the randomized controlled trial.

	Baseline (*n* = 167)		48-Hours (*n* = 154)	
Variable	HDIVC (*n* = 84)	Placebo (*n* = 83)	HDIVC (*n* = 82)	Placebo (*n* = 72)
Survived and in the ICU, *n* (%)	84 (100)	83 (100)	82 (97.6)	72 (86.7)
Age, years (mean, SD)	52.7 (17.5)	56.8 (15.7)	52.3 (17.3)	55.8 (15.7)
Men, *n* (%)	45 (53.6)	45 (54.2)	44 (53.7)	39 (54.2)
Subjects with ABG available, *n* (%)	80 (95.2)	82 (98.8)	62 (75.6)	65 (90.2)
Subjects with cfDNA available plasma, *n* (%)	84 (100)	83 (100)	81 (98.8)	70 (97.2)
Subjects with syndecan-1 available plasma, *n* (%)	83 (98.8)	83 (100)	80 (96.4)	70 (97.2)

Abbreviations: ABG, arterial blood gas; cfDNA, cell-free deoxyribonucleic acid; HDIVC, high-dose intravenous vitamin C. N, number; SD, standard deviation.

**Table 2 nutrients-14-04415-t002:** Effect of cfDNA and syndecan-1 on 28-day all-cause hospital mortality. Adjusted (multiple) logistic regression table outlining Odds Ratio (OR) of death for each incremental unit increase in baseline and 48 h change (delta, Δ) of the plasma cfDNA and syndecan-1 levels.

Predictor	Odds Ratio	Standard Error	z	*p*-Value	95% Confidence Intervals	
Baseline syndecan-1, each unit increase, ng/mL	1.3	0.1	4.9	<0.001	1.2	1.4
48 h Δsyndecan-1, each unit increase, ng/mL	1.3	0.1	3.2	0.001	1.1	1.6
Baseline cfDNA, each unit increase, ng/µL	1.1	0.2	0.5	0.650	0.8	1.5
48 h ΔcfDNA, each unit increase, ng/µL	1.8	0.5	2.1	0.035	1.0	3.0
Constant	0.0	0.0	−5.7	<0.001	0.0	0.1

## Data Availability

Not applicable.

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
