# Peer review of "Biological Effects of Intravenous Vitamin C on Neutrophil Extracellular Traps and the Endothelial Glycocalyx in Patients with Sepsis-Induced ARDS"

_nutrients, 2022, doi:10.3390/nu14204415_

Round 1

Reviewer 1 Report

Review comments: Biological Effects of Intravenous Vitamin C on Neutrophil Extracellular Traps and the Endothelial Glycocalyx in patients with sepsis-induced ARDS

This is part of a much larger study, which was not originally designed to look at these biomarkers (cell-free DNA as a marker of NET formation, and syndecan-1 as a marker of endothelial damage), and is therefore an entirely post-hoc analysis.

The rationale of the study is sound: the authors have measured biomarkers as known surrogates for the loss of endothelial integrity (syndecan-1) and large-scale neutrophil activation/inflammation (NET release i.e. cell-free DNA) which are associated with the onset of ARDS. Given the outcome of the larger study (decreased 28-day all-cause mortality with HDIVC) these are encouraging findings based in proven science. 

1.    Can the authors confirm that, in each case where statistical significance is shown as equal to xx (e.g. p = 0.05), that this is accurate? We usually use less than (P < 0.05 or P £ 0.05).

2.    Line 44: the introduction talks about the finding that HDICV reduced 28-day all-cause mortality but the ref given (#9)is only for the “reply” piece, rather than the original CITRIS-ALI study (#8), in which the actual finding was, even though it was not concluded to be a significant outcome.

3.    At no point do the authors say why they measured the cfDNA or syndecan-1 at 48h when the intervention was for 96h. There are several places in the manuscript where it refers to the effect of a 96-hour intervention on a 48h outcome.

4.    Line 58: NETs don’t actually contain neutrophil granules.

5.    Line 60: The text mentions that other studies have investigated the prognostic value of cfDNA, but it doesn’t say what they concluded.

6.    Line 63: doesn’t say why the authors hypothesise that HDIVC might reduce cfDNA or syndecan-1.

7.    Line3 71: please make it clear exactly how the patients were infused: was it a 96-hour infusion (non-stop) that delivered 50 mg/kg every 6h, or was it an infusion every 6h (over a 96-hour total) during which 50 mg/kg were given each time?  

8.    Line 90: SOFA scores are mentioned several times in the manuscript but they aren’t reported in the results: the 48h change of biomarkers was not evaluated with regard to SOFA scores in this paper (as far as I can ascertain, though I may have missed something subtle?)

9.    Line 93 on: with regard to the SOFA scores, the new conclusion in ref #9 does accept that there was a significant improvement at 96h (in the HDIVC) but not at 48h. Is there some comment or discussion about these timings?

10.Line 113: The abbreviation for PaO2/FiO2 is unnecessary, unless those data should have also been included in the table (Table 1).

11. Figure 1: why does the text talk about the mean and SD but the figure shows the median? This is incredibly unhelpful and makes it very difficult to interpret. I think that showing the mean ± SD would far better illustrate the findings.

12. Line 124: which statistical test was used? Why were there no 95% CI included for the 48-hour mean cfDNA decrease when they were included for the previous measurements?

13. Figure 1: My understanding of confidence intervals is that they should not cross the zero line and that if they do, this negates the confidence in the result. These clearly both cross zero, please explain why this approach was sued to present the data (as opposed to the mean ± SD)?

14. Fig. 1 Line 126: it doesn’t show an increase in cfDNA in the HDIVC group so the title is inaccurate.

15. Line 128: why is it valid to leave the outliers off the graph? Were they included in the data analysis?

16. Why were there no stats test done on this graph?

17. Fig. 2: again, why talk about mean ± SD but show median and CI?

18.Line 140: why are the outliers excluded again? Are they included in analyses?

19. What stats test was done on this graph?

20. Line 147: I suspect should relate to 28-day mortality and not HDIVC (copy/paste error)?

21. Forgive my ignorance but as a non-stats specialist I cannot understand how Table 2 gives any information about the effect of HDIVC on mortality. It is mentioned later in the discussion but there is no explanation of what the odds ratio actually tells us (with regard to HDIVC).

22. Line 167: is the delta sign supposed to be there (since the test also says “elevations”)?

23. Line 169: what is a normal PaO2/FiO2 ratio – give some form of context/units.

24. Fig. 4: the y axis label says P/F ratio when it should be PaO2/FiO2

25. Graph: does the single point right beside the y axis in the placebo group skew the regression line? Otherwise, the scatters look very similar, albeit more spread out in the placebo group.

26. Line 180: The HDIVC group are represented by triangles, not crosses.

27. Line 183: this study reports on the effects of a 48-hour intervention not a 96-hour one.

28. Line 186: Please explain how the HDIVC impact on syndecan-1 and cfDNA independently predict lower 28-day all-cause mortality. It isn’t obvious to me from Table 2 or the text.

29. Line 216: This study did not show an effect at 96-hours, measurements were done on the 48-hour samples.

30. Line 219: “disclosed” should probably be “demonstrated”?

31. Line 225: vitamin C is needed for appropriate neutrophil death, not their survival. Adequate ascorbate is required for PMN to undergo apoptosis (when unstimulated) and to undergo phosphatidylserine- directed clearance by macrophages once activated. NET formation may not necessarily mean PMN death (termed “vital NETosis”). The aim of giving ascorbate is not to prevent PMN death but to enable it to happen in a timely and “clean” manner to help resolve inflammation.

32. Lin3 246: why would increasing ascorbate decrease neutrophil activation? It may decrease NET formation and (therefore cfDNA) but the stimuli acting to activate PMN will still be present in sepsis/ARDS so the additional ascorbate is unlikely to prevent their activation.

Author Response

Please find attached a DOCX file with the response to Reviewer 1. 

Thank you,

Reviewer 2 Report

A well written article for which I congratulate you. I appreciated that you took into consideration elements that, being analyzed, can influence the final result. In the light of these data, I think it is worth initiating a new study that also takes into account these elements.

Author Response

We thank reviewer 2 for the kind words. 

Reviewer 3 Report

My review is limited solely to the statistical aspects.

I have compared the current version of the paper by comparing it with the previous ones and I think I can say that, in the last version of the paper, statistics are treated in a satisfactory way.

The only small observation I can make is to report the p-value values (even if not significant) in the caption of figure 4.